# Rheological and Mechanical Properties of Silica/Nitrile Butadiene Rubber Vulcanizates with Eco-Friendly Ionic Liquid

**DOI:** 10.3390/polym12112763

**Published:** 2020-11-23

**Authors:** Munir Hussain, Sohail Yasin, Hafeezullah Memon, Zhiyun Li, Xinpeng Fan, Muhammad Adnan Akram, Wanjie Wang, Yihu Song, Qiang Zheng

**Affiliations:** 1MOE Key Laboratory of Macromolecular Synthesis and Functionalization, Department of Polymer Science and Engineering, Zhejiang University, Hangzhou 310027, China; munir88@zju.edu.cn (M.H.); soh.yasin@gmail.com (S.Y.); yunlemon@zju.edu.cn (Z.L.); fxpolymer@zju.edu.cn (X.F.); m.adnanansari232@zju.edu.cn (M.A.A.); zhengqiang@zju.edu.cn (Q.Z.); 2College of Textile Science and Engineering (International Institute of Silk), Zhejiang Sci-Tech University, Hangzhou 310018, China; hm@zstu.edu.cn; 3College of Materials Science and Engineering, Henan Key Laboratory of Advanced Nylon Materials and Application, Zhengzhou University, Zhengzhou 450001, China; wwj@zzu.edu.cn

**Keywords:** nitrile butadiene rubber, rheology, ionic liquids, tensile properties

## Abstract

In this paper we designed greener rubber nanocomposites exhibiting high crosslinking density, and excellent mechanical and thermal properties, with a potential application in technical fields including high-strength and heat-resistance products. Herein 1-ethyl-3-methylimidazolium acetate ([EMIM]OAc) ionic liquid was combined with silane coupling agent to formulate the nanocomposites. The impact of [EMIM]OAc on silica dispersion in a nitrile rubber (NBR) matrix was investigated by a transmission electron microscope and scanning electron microscopy. The combined use of the ionic liquid and silane in an NBR/silica system facilitates the homogeneous dispersion of the silica volume fraction (*φ*) from 0.041 to 0.177 and enhances crosslinking density of the matrix up to three-fold in comparison with neat NBR, and also it is beneficial for solving the risks of alcohol emission and ignition during the rubber manufacturing. The introduction of ionic liquid greatly improves the mechanical strength (9.7 MPa) with respect to neat NBR vulcanizate, especially at high temperatures e.g., 100 °C. Furthermore, it impacts on rheological behaviors of the nanocomposites and tends to reduce energy dissipation for the vulcanizates under large amplitude dynamic shear deformation.

## 1. Introduction

Nitrile butadiene rubber (NBR) with exceptional oil resistance does not possess a self-reinforcing and strain-induced crystallization effect [1]. However, nanoparticles such as carbon black or silica are extensively used to reinforce NBR vulcanizates [2]. Dispersion of nanoparticles in rubbers significantly influences the linear [3,4,5] and non-linear rheological behaviors [6,7] depending on nanoparticle topology, temperature, time and interfacial interactions [8,9,10]. The nanocomposites at low frequencies undoubtedly exhibit an apparent liquid to solid transition as filler volume fraction (*φ*) increases [11,12] and, meanwhile, dynamic modulus reduces greatly with increasing strain amplitude beyond the linear viscoelastic region [13,14,15]. The former is termed the reinforcement effect and the latter is referred to as the Payne effect [16,17,18], both being long-standing topics in the fields of rheology and rubber nanocomposites.

The properties of silica-filled rubber nanocomposites rely on the rubber-silica interfacial bonding and silica dispersion [19,20]. Silica nanoparticles with high surface energy and multiple silanol groups on their surface are hard to disperse homogeneously but tend to agglomerate in rubber matrices [8,21,22]. To improve the dispersity, silica is often modified via surfactant wrapping, electron beam or plasma treatments [23,24], polymer grafting and coupling agent modification [25,26]. Among many silane coupling agents bis(*γ*-triethoxysilylpropyl)-tetrasulfide (Si69) is one of the most commonly used one to promote the silica dispersion [27,28]. Silica is used to replace petrochemical fillers to reduce the environmental impacts [29]. For better dispersity of silica in rubber matrices, high amounts of silane are required [30], which, however, brings the safety risks with ethanol emission during compounding and vulcanization. Importance of sustainability in the rubber manufacturing industry is increasing, and therefore ecofriendly and sustainable agents are needed to solve these technical and safety problems associated with rubber compounding and vulcanization.

In recent years ionic liquids (ILs) have exhibited great potential in the field of science and technologies [31] for use as clean, ecofriendly chemicals and efficient alternative resources for the reduction of volatile organic solvents. They have many expectational physical, thermal, chemical and biological properties [32]. With strong chemical and thermal stability and extremely low vapor pressure, ILs can improve colloidal stability and polymer elasticity [33]. ILs are of various types, such as pyridinium-, imidazolium-, pyrrolidinium- and ammonium and phosphonium-based, etc. [34]. Imidazolium-based ILs consist of a cationic imidazolium ring and have applications in the nuclear industry, while 1-butyl-3-methyl imidazolium chloride is used as an electrolyte to recover uranium from spent nuclear fuel [35]. For ILs used in polymer nanocomposites, it is essential to understand the variations of thermodynamics and the dynamics of polymer chains. Non-volatile, low toxicity and ecofriendly ILs exhibit great potential in rubber nanocomposite formulation [34,36,37] due to their ability to regulate the dispersions of fillers [38,39] and zinc oxide activators, the filler–rubber interfacial interactions [40] and the crosslinking density of the matrix, as well as the rheological, mechanical, optical, and thermal properties of the nanocomposites [41,42]. Phosphoniums demonstrate a catalyst role for the silanization reaction between silica and Si69 in styrene-butadiene rubber nanocomposites, which generally improves the silica dispersion and the silica–rubber interfacial interactions [21]. In silica filled butadiene rubber vulcanizates, the introduction of ILs (10 wt% of silica in weight) could effectively resolve the migration issue of rubber additives such as silica, sulphur and zinc oxide due to the improved interfacial interaction and crosslinking density in comparison with the traditional silane-incorporated vulcanizates [43].

Herein, ecofriendly 1-ethyl-3-methylimidazolium acetate ([EMIM]OAc) is used for the first time in silica-filled NBR vulcanizates for regulating the rheological and mechanical behaviors. It has vast applications in different fields and it may be used to mediate the NBR crosslinking reaction and silica dispersity. The results show enhancement in mechanical properties at both room temperature and high temperatures (60 °C and 100 °C), along with improved crosslinking density and silica dispersity. The dynamic rheological analyses illuminate the effects of [EMIM]OAc on the viscoelastic nature of NBR vulcanizates.

## 2. Materials and Methods

### 2.1. Materials

A commercially available nitrile butadiene rubber (NBR, 4155C, with 41 wt% acrylonitrile content, 355 kg/mol weight-averaged molecular weight and 0.97 g/cm^3^ density) was purchased from Lanxess, Germany. Hydrophilic fumed silica (A200) with 12 nm diameter of primary particle and 200 ± 25 m^2^ specific surface area was purchased from Evonik Degussa Co., Akron, OH, USA. Bis(*γ*-triethoxysilylpropyl) tetrasulfide (Si69) was bought from Jessica Chem. Co., Ltd., Hangzhou, China. 1-Ethyl-3-methylimidazolium acetate ([EMIM]OAc) was supplied by Linzhou Keneng Material Co., Ltd., Linzhou, China; the structure of [EMIM]OAc is shown in Figure 1. Antioxidizing agent *N*-(1,3-dimethylbuty)-*N*′-phenyl-*p*-phenylenediamine (6PPD) and vulcanizing agent sulphur were obtained from Anzhe Chem. Co., Ltd., Suzhou, China, and Zhejiang Yongjia Chem. Co., Ltd., Wenzhou, China, respectively. Accelerator *N*-cyclohexyl-2-benzothiazolyl sulfenamide (CZ) was supplied by Tokyo Chem. Ind. Co., Ltd., Tokyo, Japan. Zinc oxide and stearic acid were purchased from Aladdin Ind. Co., Shanghai, China, and Sinopharm Chem. Reagent, Shanghai, China, respectively. All the ingredients were utilized as received without further purification except for silica that was dried for 24 h in an oven under vacuum condition at 110 °C.

### 2.2. Samples Preparation

Ecofriendly compounds were prepared by mixing NBR with silica, silane, ionic liquid, zinc oxide, and other additives using an internal mixer (XSS-300, Kechuang Co., Ltd., Shanghai, China) at 40 revolutions per minute at 140 °C for 25 min. At the start of mixing, the antioxidant 6PPD was added in for preventing possible thermal oxidation and degradation throughout processing and rheological investigations. Sulphur and accelerator CZ were mixed in the compounds for 10 min at 100 °C. The additives were expressed with different parts per hundred rubber (phr), as mentioned in Table 1. Si69 or Si69/[EMIM]OAc (1/1) mixture [43] was used as coupling agent for promoting silica dispersion. For further improvement of silica dispersion, the compounds were milled additionally on a two-roll mixer (XK-160, Zhanjiang Rubber and Plastic Machinery Co., Ltd., Zhanjiang, China) at room temperature. Subsequently, the compounds were compressed on a vulcanizer (XL-25, Xinli Rubber Machinery Co., Ltd., Huzhou, China) at 10 MPa and 140 ± 5 °C for 30 min into sheets of 2.0 mm in thickness. The sheets were cut into discs of 25.0 mm in diameter for rheological testing.

### 2.3. Crosslinking Density

Crosslinking density (*v_c_*) of NBR in the nanocomposites was calculated according to Flory–Rehner equation [44,45,46]
(1)$$v_{c}=-\frac{\ln\left(1-v_{r}\right)+v_{r}+\chi_{1}v_{r}^{2}}{V\left(v_{r}^{1/3}-v_{r}/2\right)}$$
through equilibrium swelling in toluene at room temperature for 3 days. The swollen vulcanizates were dried in vacuum oven for 24 h at 60 °C. In Equation (1), *χ*_1_ = 0.6 is the Flory–Huggins interaction parameter, *V* = 106.4 cm^3^/mol is molar volume of toluene, and *v_r_* is volume fraction of rubber calculated according to:(2)$$v_{r}=\frac{\left(m_{2}-M_{s}m_{0}\right)/\rho_{r}}{\left(m_{2}-M_{s}m_{0}\right)/\rho_{r}+\left(m_{1}-m_{2}\right)/\rho_{s}}$$

Here, *m*_0_ is weight of the fresh sample, *m*_1_ and *m*_2_ are weights of the swollen and dried samples, *ρ_s_* = 0.87 g/mL and *ρ_r_* = 0.97 g/mL are densities of toluene and NBR, respectively, and *M*_s_ presents silica weight fraction in the prepared nanocomposites.

### 2.4. Transmission Electron Microscope

With the help of a transmission electron microscope (TEM, JEM-1230, JEOL, Tokyo, Japan), morphological observation was performed at an acceleration voltage of 80 kV to examine the dispersity of silica and zinc oxide. The samples were prepared from vulcanizates by freezing the microtome section.

### 2.5. Scanning Electron Micrographs

Morphology of the nanocomposites was observed by scanning electron microscopy (SEM, S-4800, Hitachi, Japan). The samples were fractured in liquid nitrogen and the fracture surfaces were sputter-coated with gold-palladium through a Denton Desk-1 vacuum sputter coater (Denton Vacuum, LLC, Moorestown, NJ, USA). Furthermore, mapping of chemical elements on the sample surface was conducted on SEM-energy dispersive spectroscopy (SEM-EDS) to check the distribution and migration of some ingredients in the vulcanizates.

### 2.6. Mechanical Properties

Uniaxial tensile tests were performed on an electronic universal testing machine with a furnace facility (Zwick roell Z020, Kennesaw, GA, USA) at room temperature, 60 °C, and 100 °C at a crosshead velocity 50 mm/min. The specimens were cut into a dog-bone shape of 12.5 mm (length) × 2 mm (width) × 2 mm (thickness) in the working section. Before conducting a high-temperature test, the samples were stored for 1 h in the furnace.

### 2.7. Rheological Measurements

Rheological measurements were performed on a strain-controlled rheometer (ARES-G2, TA Instruments, New Castle, DE, USA) with a geometry equipped serrated texture (diameter 25 mm) to prevent samples slipping, which permitted similar measurement of rheology as the use of a smooth surface plate [47]. Time sweep of the compounds was performed with a time duration of 5000 s at a strain amplitude (*γ*) 0.5% and frequency (*ω*) 10 rad/s at 140 °C. The non-linear responses of the vulcanizates were measured by *γ*-sweeps from *γ* = 0.01% to 100% at 100 °C at 10 rad/s. Before carrying out the measurements, the samples were equilibrated for 5 min. The linear responses were measured by *ω*-sweeps from *ω* = 0.0157 rad/s to 600 rad/s at *γ* = 0.03% at 100 °C. Before carrying out the measurements, the samples were equilibrated for 3 min.

## 3. Results and Discussion

### 3.1. Crosslinking Density and Vulcanization Dynamics

Commonly *v*_c_ is an important factor influencing the mechanical properties of vulcanizate nanocomposites [48]. Figure 2A shows *v_c_* of NBR in the vulcanizates. For the vulcanizates with Si69, *v_c_* increases with increasing silica volume fraction (*φ*) in a narrow range from 1.5 × 10^−4^ mol/cm^3^ to 2.8 × 10^−4^ mol/cm^3^. Once [EMIM]OAc is combined with Si69, *v*_c_ becomes higher than that using Si69 only. At *φ* = 0.177, *v_c_* even doubles. The increment of *v_c_* could be ascribed to the [EMIM]OAc promoted conversion of the silanol groups into silanolates, which avoids adsorption of vulcanizing ingredients on the surface of silica and increases the curing rate [49]. The ILs may react with the nitrile group of NBR [50] for improving the crosslinking efficiency. Furthermore, [EMIM]OAc can greatly improve dispersions of silica and zinc oxide (as will be discussed later), promote the silanization reactivity of silica, and improve the crosslinking reaction of NBR [18,39]. For the NBR gum, the induction time of vulcanization is about 200 s, and hereafter the complex modulus (*G**) increases with increasing vulcanization time, as shown in Figure 2B. For the nanocomposites, this time is shortened to about 80 s and the introduction of [EMIM]OAc does not influence the vulcanization dynamics markedly. This means that the introduction of [EMIM]OAc improves crosslinking density mainly at the very early stage of vulcanization, which is beneficial to processing and optimization of the mechanical properties of the nanocomposite.

### 3.2. Dispersion of Silica and Other Additives

Many additives are included in nanocomposite formulation and their reaction products generated during rubber vulcanization would migrate in the service of the products. In particular, the migrations of silica and curatives play a vital role in the vulcanizates, which would deteriorate the mechanical, thermal and rheological properties during service. To attain good property stability, the filler–rubber interfacial interactions should be mediated crucially [49,51] while the general methodologies for the industry are not successful until the usage of ILs [43]. Figure 3 shows photographs of the compounds and vulcanizates after being stored at room temperature for one year. Differing form the silica/butadiene rubber nanocomposites [43], there is no surface staining or migration of silica and additives during the long-term storage of both the compounds and vulcanizates of NBR that contain the polar acrylonitrile unit. However, in comparison with the nanocomposites containing Si69 only, the introduction of [EMIM]OAc changes the color obviously.

Vulcanization further causes color changes especially in the case containing [EMIM]OAc. SEM-EDS elements mapping is performed to confirm the effectiveness of ILs on rubber vulcanization and silica dispersion. Figure 4 shows elements mapping at *φ* = 0.177 (see Appendix A
Appendix A for *φ* = 0.04). The mapping shows clear staining, and migration of silica and the other additives onto the surface of the Si69-containing vulcanizates. The abundance of several elements on the surface of the Si69/[EMIM]OAc-containing vulcanizates is lower than those on the surface of the Si69-containing ones, revealing the improved dispersity of silica and the other additives during vulcanization of the nanocomposites containing [EMIM]OAc.

The distributions of silica, sulphur and zinc oxide and so on are further confirmed with quantitative EDS spectra (Figure 5). At *φ* = 0.04, the spectra are similar for the Si69- or Si69/[EMIM]OAc-containing vulcanizates. At *φ* = 0.177, the relative weight percentages of sulphur and zinc elements (2.85% and 3.48%) on the surface of Si69-containing vulcanizates are higher than those (2.41% and 0.09%) of the Si69/[EMIM]OAc-containing vulcanizates, revealing the significantly improved dispersion of sulphur and zinc oxide with the aid of [EMIM]OAc [52,53].

The improved dispersions of sulphur and zinc oxide enhance the vulcanization activity and are responsible for the higher *v_c_* of the Si69/[EMIM]OAc-containing vulcanizates in comparison with that of the Si69-containing ones (Figure 2A). TEM is used to evaluate the dispersions of silica and zinc oxide in the vulcanizates, as shown in Figure 6. The Si69-containing vulcanizates have few large silica agglomerates and big zinc oxide particles owing to the hydrophilic surface and low efficiency of silanization [54]. In comparison, the incorporation of Si69 and [EMIM]OAc could improve silica dispersion notably and reduce the size of zinc oxide particles efficiently. The alkaline [EMIM]OAc is expected to improve the catalytic efficiency for silanization of silica to improve the wettability of silica to NBR chains [55]. On the other hand, the zinc oxide particles with greatly reduced size exhibit improved activation to the rubber vulcanization. The modified silica dispersity and rubber vulcanization yield in mediation of mechanical properties of the vulcanizates, as will be discussed in the next section.

### 3.3. Thermal Properties

Glass transition temperature (*T_g_*) of NBR in the vulcanizates at *φ* = 0.177 was estimated at the middle point of heat capacity jump measured by modulated differential scanning calorimetry at a heating rate of 1 °C/min [56,57,58] (Appendix A). In the absence of the coupling agent, *T_g_* of the filled vulcanizate is the same as that of the gum vulcanizate (Appendix A). By addition of Si69 or Si69/[EMIM]OAc, *T_g_* elevates by 0.8 °C and 1.5 °C, respectively, attributed to the improved silica–rubber interfacial interaction [59,60]. The results suggest that [EMIM]OAc is more effective than Si69 for the creation of a strongly interacting silica–rubber interface. Thermal stability is vital in application of vulcanizates [61]. According to derivative thermogravimetry curves (Appendix A), the gum vulcanizate in nitrogen atmosphere shows decomposition peaks at 200 °C and 383 °C, respectively. At *φ* = 0.177, the rate of the low-temperature decomposition is reduced considerably and the high-temperature decomposition peak is shifted to 409 °C and 412 °C, respectively, when Si69 or Si69/[EMIM]OAc is used as a coupling agent. The improved thermal stability in the presence of [EMIM]OAc is associated with interfacial interaction that stabilizes NBR chains by blocking the chain ends with silica [57,62].

### 3.4. Tensile Properties

The mechanical properties of rubber nanocomposites usually vary with temperature and may deteriorate greatly at relatively high temperatures. To examine the influence of temperature, tensile stress-strain behaviors at room temperature, 60 °C, and 100 °C were measured (Appendix A) [63,64] and tensile strength, strain at break and stresses at 100% and 300% strains are determined. The results are listed in Table 2. Firstly, tensile strength, strain at break, and stresses at 100% and 300% strains decrease with temperature due to the weakened intermolecular interactions. Secondly, tensile strength and stresses at 100% and 300% strains increase with increasing *φ*, which is ascribed to the reinforcement effect and improved crosslinking density induced by the filler (Figure 2A). Thirdly, tensile strength and stresses at 100% and 300% strains of the Si69/[EMIM]OAc-containing vulcanizates are higher than those of the Si69-containing ones, which is ascribed to the modifications of filler dispersion and vulcanization efficiency with the aid of ILs [65,66,67]. The Si69/[EMIM]OAc-containing vulcanizate at *φ* = 0.177 exhibits tensile strength of 31 MPa at room temperature and 11 MPa at 100 °C, signifying a potential application in high-strength and heat-resistance products. The surprisingly high strength at 100 °C endowed by ILs might be ascribed to the improved crosslinking efficiency (Figure 2A) and thermal stability (Appendix A).

### 3.5. Rheological Behaviours

Rheological responses of the vulcanizates are dependent on the composition. Figure 7 shows the linear rheological response at *γ* = 0.03%. Storage modulus (*G′*), loss modulus (*G″*) and loss tangent (tan *δ*) of the gum vulcanizate are weakly dependent on frequency (ω) [68], which is ascribed to the relaxation of a non-ideally crosslinked molecular network. Filling in general elevates both *G*’ and *G*’’ except for a reduction of *G*’’ at high *ω* for the nanocomposites at *φ* = 0.041. The introduction of silica at *φ* = 0.041 greatly reduces tan *δ* and the effect is more marked in the presence of Si69/[EMIM]OAc than Si69 only as the coupling agent. With increasing *φ*, tan *δ* of the vulcanizates increases due to the relaxation associated with interparticle and filler–rubber interfacial frictions. The comparison of rheological data at *ω* = 120 rad/s is shown in Figure 8. It is seen that the presence of Si69/[EMIM]OAc lowers *G*’, *G*’’ and tan *δ* in comparison with the case of Si69 as a coupling agent, except for *G’* at *φ* = 0.177. The tan δ reduction is consistent with the improved filler dispersion and crosslinking efficiency, and is beneficial for alleviating the dissipation of mechanical energy during dynamic deformations.

Non-linear rheological responses are shown in Figure 9 for the gum and filled vulcanizates at 100 °C at 10 rad/s. At low *γ* all the vulcanizates present a linear viscoelastic region where *G’*, *G*’’, and tan *δ* remain near constant. The linear viscoelastic region is shortened with increasing *φ*, which could be ascribed to the strain amplification effect arising from the filler [68,69,70]. Beyond the linear viscoelastic region, *G’* decreases and tan *δ* increases rapidly with increasing *γ*, which is called the Payne effect [71,72]. Accompanying this effect, *G’’* of the gum vulcanizate exhibits a weak strain overshoot [57,73] indicating the formation of a non-ideal crosslinking network containing many defects like dangling chains [47] that strengthen the intermolecular friction during dynamic deformation.

Filling causes the overshoot to become broad and gradually weakened, which could be ascribed to the adsorption of chains on the surface of silica nanoparticles. In comparison with Si69, the usage of Si69/[EMIM]OAc as coupling agent reduces tan *δ* in the linear viscoelastic region but it does not influence the increment of tan *δ* in the non-linear regime. This means that the Payne effect is not only dependent on the filler dispersity but also on the network of the rubber matrix.

## 4. Conclusions

In comparison with Si69, a combined usage of Si69 and [EMIM]OAc in a weight ratio of 1/1 in silica filled NBR vulcanizates can improve the crosslinking degree and filler dispersity, resulting in an improvement in tensile strength especially at high temperatures (60 °C and 100 °C). The presence of [EMIM]OAc could decline *G*’’ and tan *δ* in the linear viscoelastic region, indicating a reduction of energy dissipation. Furthermore, the Payne effect does not alter markedly by introduction of [EMIM]OAc, suggesting that this effect involves both the filler dispersity and crosslinking density. The results are helpful for designing greener rubber nanocomposites with optimized crosslinking density and filler dispersity and for reducing safety risks associated with alcohol emission due to the usage of silane during rubber compounding.

## Figures and Tables

**Figure 1 polymers-12-02763-f001:**
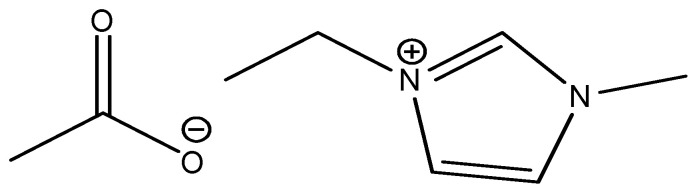
Structure of 1-ethyl-3-methylimidazolium acetate.

**Figure 2 polymers-12-02763-f002:**
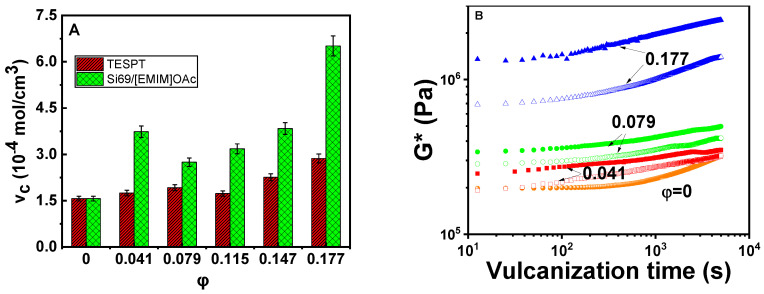
(**A**) Crosslinking density (*v*_c_) of nitrile butadiene rubber (NBR) in the vulcanizates and (**B**) complex modulus (*G**) of the nanocomposites containing bis(*γ*-triethoxysilylpropyl) tetrasulfide (Si69) (hollow symbols) or Si69/1-ethyl-3-methylimidazolium acetate ([EMIM]OAc) coupling agent (solid symbols) against vulcanization time.

**Figure 3 polymers-12-02763-f003:**
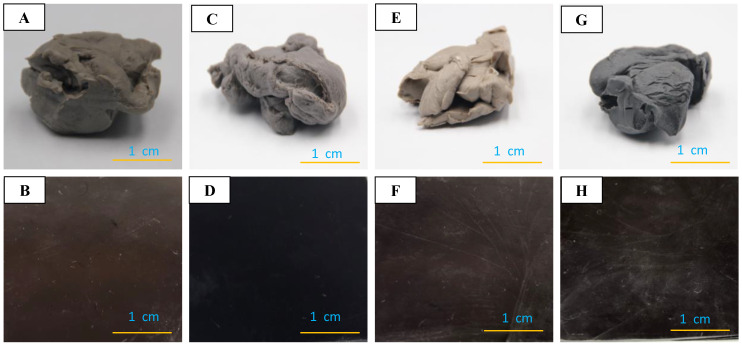
Photographs of one-year-stored compounds (**A**,**C**,**E** and **G**) and vulcanizates (**B**,**D**,**F** and **H**) comprising Si69 (**A**,**B**,**E** and **F**) or Si69/[EMIM]OAc (**C**,**D**,**G** and **H**) at *φ* = 0.04 (**A**–**D**) and *φ* = 0.177 (**E**–**H**). Color differences are observed for the compounds and vulcanizates and the marks on the surface of the vulcanizates are generated after separation of the mold for the compression molding.

**Figure 4 polymers-12-02763-f004:**
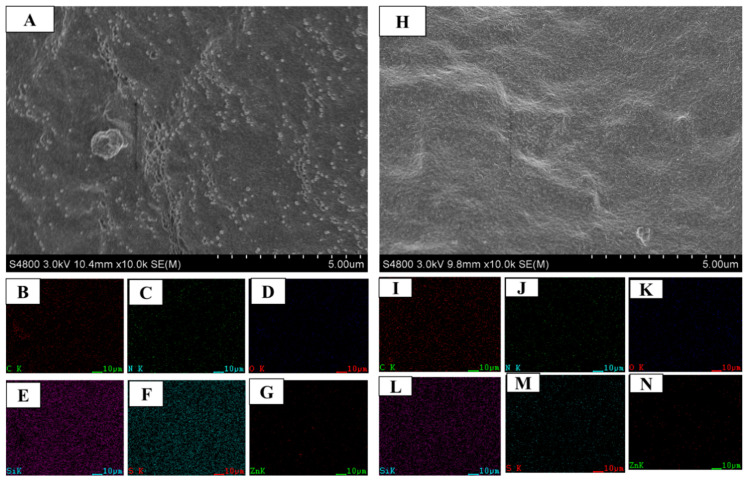
Scanning electron microscope (SEM) images of the vulcanizates (φ = 0.177) containing Si69 (**A**) or Si69/[EMIM]OAc (**H**) and SEM-energy dispersive spectroscopy (SEM-EDS) maps of carbon (**B** and **I**), nitrogen (**C** and **J**), oxygen (**D** and **K**), silicon (**E** and **L**), sulphur (**F** and **M**), and zinc elements (**G** and **N**).

**Figure 5 polymers-12-02763-f005:**
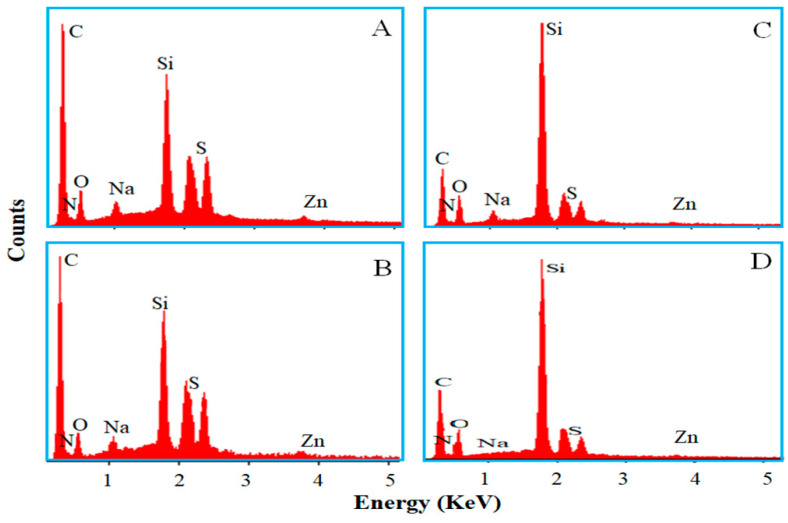
Energy-dispersive spectroscopy (EDS) spectra of the Si69- (**A** and **C**) and Si69/[EMIM]OAc-containing vulcanizates (**B** and **D**) at *φ* = 0.04 (**A** and **B**) and *φ* = 0.177 (**C** and **D**).

**Figure 6 polymers-12-02763-f006:**
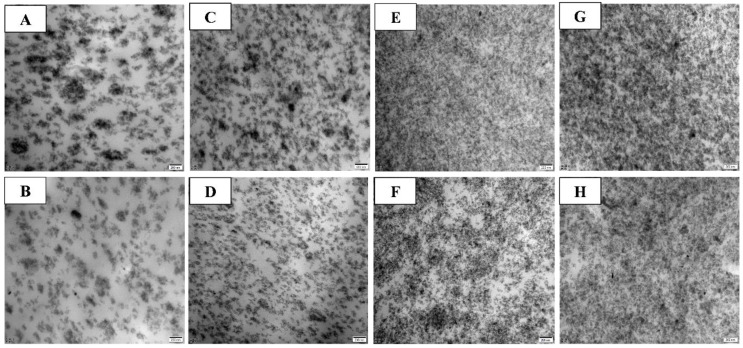
Transmission electron microscopy (TEM) images of the Si69- (**A**,**C**,**E** and **G**) or Si69/[EMIM]OAc-containing vulcanizates (**B**,**D**,**F** and **H**) at *φ* = 0.041 (**A** and **B**), 0.079 (**C** and **D**), 0.147 (**E** and **F**), and 0.177 (**G** and **H**).

**Figure 7 polymers-12-02763-f007:**
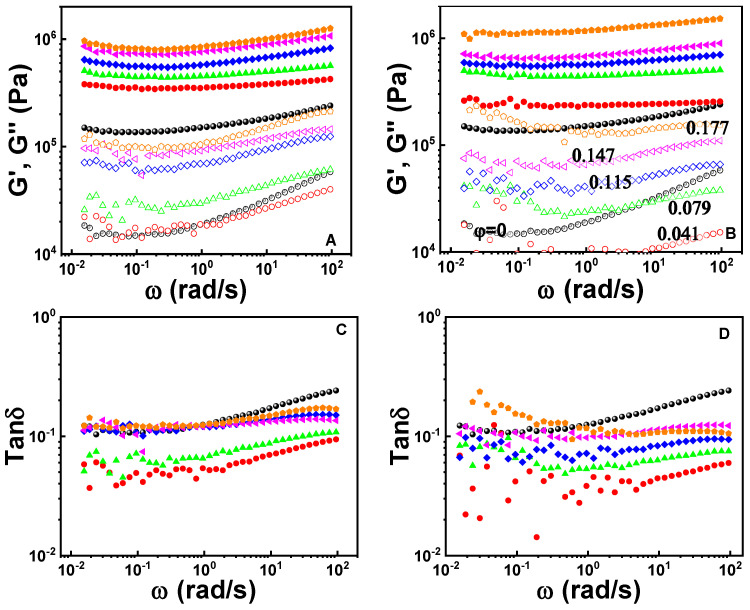
Storage modulus (*G*’, solid symbols), loss modulus (*G*’’, hollow symbols) and loss tangent (tan *δ*) as a function of frequency (*ω*) for the gum and filled vulcanizates containing Si69 (**A** and **C**) or Si69/[EMIM]OAc (**B** and **D**) at strain amplitude (*γ*) 0.03%.

**Figure 8 polymers-12-02763-f008:**
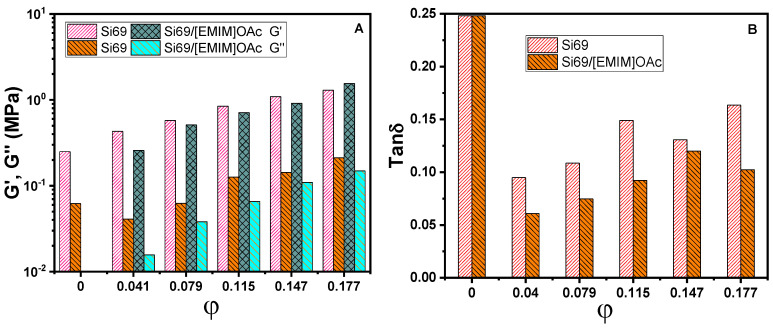
*G*’, *G*’’ (**A**) and tan *δ* (**B**) as a function of *φ* at *ω* = 120 rad/s and *γ* = 0.03% for the gum vulcanizate and its Si69–or Si69/[EMIM]OAc-containing nanocomposites.

**Figure 9 polymers-12-02763-f009:**
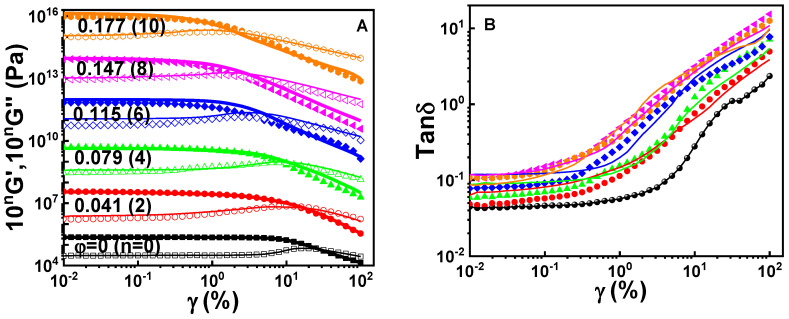
*G*’ (solid symbols and thick curves, **A**), *G*’’ (hollow symbols and thin curves, **A**) and tan *δ* (**B**) as a function of *γ* for the gum and filled vulcanizates containing Si69 (curves) or Si69/[EMIM]OAc (symbols) at 100 °C and 10 rad/s. The data of *G*’ and *G*’’ are vertically shifted by a factor of 10*^n^* as indicated.

**Table 1 polymers-12-02763-t001:** Compositions of the nanocomposites.

Ingredients	Amounts (phr) ^a^
Nitrile butadiene rubber (NBR)	100
Silica	0, 10, 20, 30, 40, 50
Si69 ^b^ or Si69/[EMIM]OAc ^c^ (1/1)	0, 0.46, 0.83, 1.15, 1.43, 1.67 (10 wt% of silica)
Antioxidant 6PPD ^d^	1
Zinc oxide	2
Steric acid	2
Sulphur	2
Accelerator CZ ^e^	1

^a^ Parts per hundred rubber (phr), ^b^ Bis(γ-triethoxysilylpropyl) tetrasulfide (Si69), ^c^ 1-Ethyl-3-methylimidazolium acetate ([EMIM]OAc), ^d^ N-(1,3-dimethylbuty)-N′-phenyl-p-phenylenediamine (6PPD), ^e^ N-cyclohexyl-2-benzothiazolyl sulfenamide (CZ).

**Table 2 polymers-12-02763-t002:** Mechanical properties of NBR vulcanizate and its Si69–or Si69/[EMIM]OAc-containing nanocomposites.

Coupling Agent		/	Si69			Si69/[EMIM]OAc		
*φ*		0	0.041	0.079	0.177	0.041	0.079	0.177
Strain at break (%)	100 °C	409 ± 143	409 ± 143	409 ± 143	409 ± 143	409 ± 143	409 ± 143	409 ± 143
	60 °C	572 ± 19	357 ± 19	369 ± 5	600 ± 43	252 ± 19	360 ± 32	563 ± 12
	RT	659 ± 48	540 ± 31	614 ± 18	566 ± 82	472 ± 19	512 ± 32	559 ± 47
Tensile strength (MPa)	100 °C	1.5 ± 0.2	2.4 ± 0.3	4.1 ± 0.2	8.5 ± 0.7	2.7 ± 0.1	5.0 ± 0.3	11.2 ± 1.0
	60 °C	2.5 ± 0.3	4.0 ± 0.3	5.1 ± 0.2	14.4 ± 1.3	4.3 ± 0.4	9.8 ± 1.6	19.6 ± 1.5
	RT	6.4 ± 0.8	14.1 ± 2.1	20.9 ± 0.9	25.5 ± 1.5	17.4 ± 1.9	25.1 ± 1.8	31.0 ± 1.3
Stress at 100% strain (MPa)	100 °C	0.89 ± 0.1	1.5 ± 0.1	1.6 ± 0.1	2.0 ± 0.1	2.36 ± 0.2	2.3 ± 0.2	3.2 ± 0.5
	60 °C	0.59 ± 0.09	1.24 ± 0.09	1.42 ± 0.1	2.42 ± 0.23	1.73 ± 0.24	2.15 ± 0.30	2.56 ± 0.20
	RT	0.85 ± 0.10	1.48 ± 0.14	1.97 ± 0.2	2.84 ± 0.16	2.22 ± 0.7	2.97 ± 0.6	4.22 ± 0.8
Stress at 300% strain (MPa)	100 °C	1.61 ± 0.16	2.64 ± 0.43	4.31 ± 0.14	6.23 ± 0.34	2.85 ± 0.42	5.37 ± 0.34	9.90 ± 0.70
	60 °C	1.43 ± 0.20	3.22 ± 0.30	4.14 ± 0.17	7.08 ± 0.46	4.67 ± 0.60	8.05 ± 0.80	8.64 ± 0.50
	RT	1.92 ± 0.30	4.58 ± 0.40	6.69 ± 0.34	11.2 ± 0.8	8.0 ± 0.2	11.5 ± 1.1	14.8 ± 0.9

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
