# Peer review of "Rheological and Mechanical Properties of Silica/Nitrile Butadiene Rubber Vulcanizates with Eco-Friendly Ionic Liquid"

_polymers, 2020, doi:10.3390/polym12112763_

Round 1

Reviewer 1 Report

The authors have used ionic liquids combined with silanizstion agent in silica filled nitrile- butadiene rubber. The influence of the ionic liquid was evaluated in tests of vulcanization and improvement in cross-linking was observed, which was evaluated by the rheological methods.

The manuscript is well-structured and presented, but overall spell-check and English-style editing is adviced prior publication. 

The manuscript specific questions are divided into overall questions (oc) constuctive critique (cc) and remarks (r):

oc:

1)  Table two contains three temperatures, 100, 60 and RT. Why were they chosen and how does it relate to the vulcanization process?

2) In the Fig. 2. the sparks are visible, what are those?

3) Can you please comment on the reactivity of IL with butadiene, especially at higher temperatures? Was there any degradation visible?

4) Were the samples sputter-coated for sake of SEM images? If not, what was the accelerating voltage? For samples that easily undergo charging, especially silica with rather large band-gap, the parameters used for imaging may aid other researchers in pursuing similar experiments. 

cc:

1) for sake of the scientific soundness please provide a scale-bar in the photographs in Fig. 2. 

r) 

1) many oddly formulated sentences appear accross the paper, e.g. "In recent years applications of ionic liquids (ILs) are increasing in different fields of science [28] and are feasible substitutes of toxic, highly flammable, hazardous, and volatile solvents used for gels, dispersions, polymeric melts and composites. " Please revise the manuscript and spell-check the sentences.

Reviewer 2 Report

The manuscript presented by Hussain et al shows an interesting NBR recipe using an ionic liquid to improve the dispersion of silica, thereby enhancing the mechanical/thermal properties of the composites therein produced. I find the manuscript very attractive for the scientific community dealing with rubber formulations, especially those dedicated to pursuing greener and less polluting materials. The results are also novel and worth publising. I think the manuscript is ready to be accepted in Polymers after some minor revisions, which are aiming to improve the impact of the work:

1) Abstract, lines 17-20: I suggest revising the sentence, it is slightly confusing. I suggest also adding a conclusive statement in the abstract, for instance explaining the contribution of this work to the field and the potentials of the materials therein produced. 

2) I suggest adding some experimental values in the abstract (e.g. line 22), and in the introduction (e.g. line 74). For instance, reflect upon how much silica was dispersed, by how much was the crosslinking density changed compared to the reference material, etc.

3) Line 24, change "like" for "e.g"

4) Line 39, when explaining the Payne effect (including hysteresis effects in rubber composites aka Müllin effect), a reference is missing. I suggest the following references: 

https://doi.org/10.1021/ma8014728

https://doi.org/10.3390/polym11122122

https://doi.org/10.1021/acsomega.8b03630

5) Introduction, line 68: please state how much is referred to as "small amount".

6) For helping the reader, I suggest adding the chemical structure of (EMIM)OAc and also a motivation for using this ionic liquid compared to other alternatives. 

7) The SEM images are taken from the cross-section or the top-surface of the samples? 

8) Figure 1: are there any standard deviations to report in the figure?

9) Please add more clear images of B, D, F, H.

10) Figure 3: I suggest increasing the size of the SEM images (A and I).

11) Line 239: I suggest describing the TGA experimental description in the main text also.  

12) Line 258: Please state how are the presented results compared to other similar studies? I also suggest adding a tensile test representative curves, at least as a supplementary figure. 

13) Line 306: I suggest adding "designing greener/less polluting rubber nanocomposites"
